# Amortized Inference Regularization

**Rui Shu**
Stanford University
ruishu@stanford.edu

**Hung H. Bui**
DeepMind
buih@google.com

**Shengjia Zhao**
Stanford University
sjzhao@stanford.edu

**Mykel J. Kochenderfer**
Stanford University
mykel@stanford.edu

**Stefano Ermon**
Stanford University
ermon@cs.stanford.edu

## Abstract

The variational autoencoder (VAE) is a popular model for density estimation and representation learning. Canonically, the variational principle suggests to prefer an expressive inference model so that the variational approximation is accurate. However, it is often overlooked that an overly-expressive inference model can be detrimental to the test set performance of both the amortized posterior approximator and, more importantly, the generative density estimator. In this paper, we leverage the fact that VAEs rely on *amortized* inference and propose techniques for *amortized inference regularization* (AIR) that control the smoothness of the inference model. We demonstrate that, by applying AIR, it is possible to improve VAE generalization on both inference and generative performance. Our paper challenges the belief that amortized inference is simply a mechanism for approximating maximum likelihood training and illustrates that regularization of the amortization family provides a new direction for understanding and improving generalization in VAEs.

## 1 Introduction

Variational autoencoders are a class of generative models with widespread applications in density estimation, semi-supervised learning, and representation learning [1, 2, 3, 4]. A popular approach for the training of such models is to maximize the log-likelihood of the training data. However, maximum likelihood is often intractable due to the presence of latent variables. Variational Bayes resolves this issue by constructing a tractable lower bound of the log-likelihood and maximizing the lower bound instead. Classically, Variational Bayes introduces per-sample approximate proposal distributions that need to be optimized using a process called variational inference. However, per-sample optimization incurs a high computational cost. A key contribution of the variational autoencoding framework is the observation that the cost of variational inference can be amortized by using an amortized inference model that learns an efficient mapping from samples to proposal distributions. This perspective portrays amortized inference as a tool for efficiently approximating maximum likelihood training. Many techniques have since been proposed to expand the expressivity of the amortized inference model in order to better approximate maximum likelihood training [5, 6, 7, 8].

In this paper, we challenge the conventional role that amortized inference plays in variational autoencoders. For datasets where the generative model is prone to overfitting, we show that having an amortized inference model actually provides a new and effective way to regularize maximum likelihood training. Rather than making the amortized inference model more expressive, we propose instead to restrict the capacity of the amortization family. Through amortized inference regularization (AIR), we show that it is possible to reduce the inference gap and increase the log-likelihood performance on the test set. We propose several techniques for AIR and provide extensive theoretical and empirical analyses of our proposed techniques when applied to the variational autoencoder and the

importance-weighted autoencoder. By rethinking the role of the amortized inference model, amortized inference regularization provides a new direction for studying and improving the generalization performance of latent variable models.

## 2 Background and Notation

### 2.1 Variational Inference and the Evidence Lower Bound

Consider a joint distribution $p_\theta(x, z)$ parameterized by $\theta$, where $x \in \mathcal{X}$ is observed and $z \in \mathcal{Z}$ is latent. Given a uniform distribution $\hat{p}(x)$ over the dataset $\mathcal{D} = \{x^{(i)}\}$, maximum likelihood estimation performs model selection using the objective

$$\max_\theta \mathbb{E}_{\hat{p}(x)} \ln p_\theta(x) = \max_\theta \mathbb{E}_{\hat{p}(x)} \ln \int_z p_\theta(x, z) dz. \tag{1}$$

However, marginalization of the latent variable is often intractable; to address this issue, it is common to employ the variational principle to maximize the following lower bound

$$\max_\theta \mathbb{E}_{\hat{p}(x)} \left[ \ln p_\theta(x) - \min_{q \in \mathcal{Q}} D(q(z) \parallel p_\theta(z \mid x)) \right] = \max_\theta \mathbb{E}_{\hat{p}(x)} \left[ \max_{q \in \mathcal{Q}} \mathbb{E}_{q(z)} \ln \frac{p_\theta(x, z)}{q(z)} \right], \tag{2}$$

where $D$ is the Kullback-Leibler divergence and $\mathcal{Q}$ is a variational family. This lower bound, commonly called the evidence lower bound (ELBO), converts log-likelihood estimation into a tractable optimization problem. Since the lower bound holds for any $q$, the variational family $\mathcal{Q}$ can be chosen to ensure that $q(z)$ is easily computable, and the lower bound is optimized to select the best proposal distribution $q_x^*(z)$ for each $x \in \mathcal{D}$.

### 2.2 Amortization and Variational Autoencoders

[1, 9] proposed to construct $p(x \mid z)$ using a parametric function $g_\theta \in \mathcal{G}(\mathcal{P}) : \mathcal{Z} \to \mathcal{P}$, where $\mathcal{P}$ is some family of distributions over $x$, and $\mathcal{G}$ is a family of functions indexed by parameters $\theta$. To expedite training, they observed that it is possible to amortize the computational cost of variational inference by framing the per-sample optimization process as a *regression* problem; rather than solving for the optimal proposal $q_x^*(z)$ directly, they instead use a recognition model $f_\phi \in \mathcal{F}(\mathcal{Q}) : \mathcal{X} \to \mathcal{Q}$ to predict $q_x^*(z)$. The functions $(f_\phi, g_\theta)$ can be concisely represented as conditional distributions, where

$$p_\theta(x \mid z) = g_\theta(z)(x) \tag{3}$$
$$q_\phi(z \mid x) = f_\phi(x)(z). \tag{4}$$

The use of amortized inference yields the variational autoencoder, which is trained to maximize the variational autoencoder objective

$$\max_{\theta, \phi} \mathbb{E}_{\hat{p}(x)} \left[ \mathbb{E}_{q_\phi(z \mid x)} \ln \frac{p(z) p_\theta(x \mid z)}{q_\phi(z \mid x)} \right] = \max_{f \in \mathcal{F}(\mathcal{Q}), g \in \mathcal{G}(\mathcal{P})} \mathbb{E}_{\hat{p}(x)} \left[ \mathbb{E}_{z \sim f(x)} \ln \frac{p(z) g(z)(x)}{f(x)(z)} \right]. \tag{5}$$

We omit the dependency of $(p(z), g)$ on $\theta$ and $f$ on $\phi$ for notational simplicity. In addition to the typical presentation of the variational autoencoder objective (LHS), we also show an alternative formulation (RHS) that reveals the influence of the model capacities $\mathcal{F}, \mathcal{G}$ and distribution family capacities $\mathcal{Q}, \mathcal{P}$ on the objective function. In this paper, we use $(q_\phi, f)$ interchangeably, depending on the choice of emphasis. To highlight the relationship between the ELBO in Eq. (2) and the standard variational autoencoder objective in Eq. (5), we shall also refer to the latter as the amortized ELBO.

### 2.3 Amortized Inference Suboptimality

For a fixed generative model, the optimal unamortized and amortized inference models are

$$q_x^* = \arg\max_{q \in \mathcal{Q}} \mathbb{E}_{q(z)} \left[ \ln \frac{p_\theta(x, z)}{q(z)} \right], \text{for each } x \in \mathcal{D} \tag{6}$$

$$f^* = \arg\max_{f \in \mathcal{F}} \mathbb{E}_{\hat{p}(x)} \left[ \mathbb{E}_{z \sim f(x)} \ln \frac{p_\theta(x, z)}{f(x)(z)} \right]. \tag{7}$$

A notable consequence of using an amortization family to approximate variational inference is that Eq. (5) is a lower bound of Eq. (2). This naturally raises the question of whether the learned inference model can accurately approximate the mapping $x \mapsto q_x^*(z)$. To address this question, [10] defined the inference, approximation, and amortization gaps as

$$\Delta_{\text{inf}}(\hat{p}) = \mathbb{E}_{\hat{p}(x)} D(f^*(x) \parallel p_\theta(z \mid x)) \tag{8}$$

$$\Delta_{\text{ap}}(\hat{p}) = \mathbb{E}_{\hat{p}(x)} D(q_x^*(z) \parallel p_\theta(z \mid x)) \tag{9}$$

$$\Delta_{\text{am}}(\hat{p}) = \Delta_{\text{inf}}(\hat{p}) - \Delta_{\text{ap}}(\hat{p}), \tag{10}$$

Studies have found that the inference gap is non-negligible [11] and primarily attributable to the presence of a large amortization gap [10].

The amortization gap raises two critical considerations. On the one hand, we wish to reduce the training amortization gap $\Delta_{\text{am}}(\hat{p}_{\text{train}})$. If the family $\mathcal{F}$ is too low in capacity, then it is unable to approximate $x \mapsto q_x^*$ and will thus increase the amortization gap. Motivated by this perspective, [5, 12] proposed to reduce the training amortization gap by performing stochastic variational inference on top of amortized inference. In this paper, we take the opposing perspective that an over-expressive $\mathcal{F}$ hurts generalization (see Appendix A) and that restricting the capacity of $\mathcal{F}$ is a form of regularization that can prevent both the inference *and* generative models from overfitting to the training set.

## 3    Amortized Inference Regularization in Variational Autoencoders

Many methods have been proposed to expand the variational and amortization families in order to better approximate maximum likelihood training [5, 6, 7, 8, 13, 14]. We argue, however, that achieving a better approximation to maximum likelihood training is not necessarily the best training objective, even if the end goal is test set density estimation. In general, it may be beneficial to regularize the maximum likelihood training objective.

Importantly, we observe that the evidence lower bound in Eq. (2) admits a natural interpretation as implicitly regularizing maximum likelihood training

$$\max_\theta \Big( \overbrace{\mathbb{E}_{\hat{p}(x)} \left[ \ln p_\theta(x) \right]}^{\text{log-likelihood}} - \overbrace{\mathbb{E}_{\hat{p}(x)} \min_{q \in \mathcal{Q}} D(q(z) \parallel p_\theta(z \mid x))}^{\text{regularizer } R(\theta; \mathcal{Q})} \Big). \tag{11}$$

This formulation exposes the ELBO as a *data-dependent regularized* maximum likelihood objective. For infinite capacity $\mathcal{Q}$, $R(\theta; \mathcal{Q})$ is zero for all $\theta \in \Theta$, and the objective reduces to maximum likelihood. When $\mathcal{Q}$ is the set of Gaussian distributions (as is the case in the standard VAE), then $R(\theta; \mathcal{Q})$ is zero only if $p_\theta(z \mid x)$ is Gaussian for all $x \in \mathcal{D}$. In other words, a Gaussian variational family regularizes the true posterior $p_\theta(z \mid x)$ toward being Gaussian [10]. Careful selection of the variational family to encourage $p_\theta(z \mid x)$ to adopt certain properties (e.g. unimodality, fully-factorized posterior, etc.) can thus be considered a special case of *posterior regularization* [15, 16].

Unlike traditional variational techniques, the variational autoencoder introduces an amortized inference model $f \in \mathcal{F}$ and thus a new source of posterior regularization.

$$\max_\theta \Big( \overbrace{\mathbb{E}_{\hat{p}(x)} \left[ \ln p_\theta(x) \right]}^{\text{log-likelihood}} - \overbrace{\min_{f \in \mathcal{F}(\mathcal{Q})} \mathbb{E}_{\hat{p}(x)} \left[ D(f(x) \parallel p_\theta(z \mid x)) \right]}^{\text{regularizer } R(\theta; \mathcal{Q}, \mathcal{F})} \Big). \tag{12}$$

In contrast to unamortized variational inference, the introduction of the amortization family $\mathcal{F}$ forces the inference model to consider the *global structure* of how $\mathcal{X}$ maps to $\mathcal{Q}$. We thus define *amortized inference regularization* as the strategy of restricting the inference model capacity $\mathcal{F}$ to satisfy certain desiderata. In this paper, we explore a special case of AIR where a candidate model $f \in \mathcal{F}$ is penalized if it is not sufficiently smooth. We propose two models that encourage inference model smoothness and demonstrate that they can reduce the inference gap and increase log-likelihood on the test set.

### 3.1    Denoising Variational Autoencoder

In this section, we propose using random perturbation training for amortized inference regularization. The resulting model—the denoising variational autoencoder (DVAE)—modifies the variational

autoencoder objective by injecting $\varepsilon$ noise into the inference model

$$\max_\theta \left( \mathbb{E}_{\hat{p}(x)} \left[ \ln p_\theta(x) \right] - \min_{f \in \mathcal{F}(\mathcal{Q})} \mathbb{E}_{\hat{p}(x)} \mathbb{E}_\varepsilon \left[ D(f(x + \varepsilon) \parallel p_\theta(z \mid x)) \right] \right). \tag{13}$$

Note that the noise term only appears in the regularizer term. We consider the case of zero-mean isotropic Gaussian noise $\varepsilon \sim \mathcal{N}(\mathbf{0}, \sigma\mathbf{I})$ and denote the denoising regularizer as $R(\theta\,;\sigma)$. At this point, we note that the DVAE was first described in [17]. However, our treatment of DVAE differs from [17]'s in both theoretical analysis and underlying motivation. We found that [17] incorrectly stated the tightness of the DVAE variational lower bound (see Appendix B). In contrast, our analysis demonstrates that the denoising objective smooths the inference model and necessarily lower bounds the original variational autoencoder objective (see Theorem 1 and Proposition 1).

We now show that 1) the optimal DVAE amortized inference model is a kernel regression model and that 2) the variance of the noise $\varepsilon$ controls the smoothness of the optimal inference model.

**Lemma 1.** *For fixed $(\theta, \sigma, \mathcal{Q})$ and infinite capacity $\mathcal{F}$, the inference model that optimizes the DVAE objective in Eq. (13) is the kernel regression model*

$$f_\sigma^*(x) = \arg\min_{q \in \mathcal{Q}} \sum_{i=1}^n w_\sigma(x, x^{(i)}) \cdot D(q(z) \parallel p_\theta(z \mid x^{(i)})), \tag{14}$$

*where $w_\sigma(x, x^{(i)}) = \frac{K_\sigma(x, x^{(i)})}{\sum_j K_\sigma(x, x^{(j)})}$ and $K_\sigma(x, y) = \exp\left( -\frac{\|x - y\|}{2\sigma^2} \right)$ is the RBF kernel.*

Lemma 1 shows that the optimal denoising inference model $f_\sigma^*$ is dependent on the noise level $\sigma$. The output of $f_\sigma^*(x)$ is the proposal distribution that minimizes the weighted Kullback-Leibler (KL) divergence from $f_\sigma^*(x)$ to each $p_\theta(z \mid x^{(i)})$, where the weighting $w_\sigma(x, x^{(i)})$ depends on the distance $\|x - x^{(i)}\|$ and the bandwidth $\sigma$. When $\sigma > 0$, the amortized inference model forces neighboring points $(x^{(i)}, x^{(j)})$ to have similar proposal distributions. Note that as $\sigma$ increases, $w_\sigma(x, x^{(i)}) \to \frac{1}{n}$, where $n$ is the number of training samples. Controlling $\sigma$ thus modulates the smoothness of $f_\sigma^*$ (we say that $f_\sigma^*$ is smooth if it maps similar inputs to similar outputs under some suitable measure of similarity). Intuitively, the denoising regularizer $R(\theta\,;\sigma)$ approximates the true posteriors with a "$\sigma$-smoothed" inference model and penalizes generative models whose posteriors cannot easily be approximated by such an inference model. This intuition is formalized in Theorem 1.

**Theorem 1.** *Let $\mathcal{Q}$ be a minimal exponential family with corresponding natural parameter space $\Omega$. With a slight abuse of notation, consider $f \in \mathcal{F} : \mathcal{X} \to \Omega$. Under the simplifying assumption that $p_\theta(z \mid x^{(i)})$ is contained within $\mathcal{Q}$ and parameterized by $\eta^{(i)} \in \Omega$, and that $\mathcal{F}$ has infinite capacity, then the optimal inference model in Lemma 1 returns $f_\sigma^*(x) = \eta \in \Omega$, where*

$$\eta = \sum_{i=1}^n w_\sigma(x, x^{(i)}) \cdot \eta^{(i)} \tag{15}$$

*and Lipschitz constant of $f_\sigma^*$ is bounded by $O(1/\sigma^2)$.*

We wish to address Theorem 1's assumption that the true posteriors lie in the variational family. Note that for sufficiently large exponential families, this assumption is likely to hold. But even in the case where the variational family is Gaussian (a relatively small exponential family), the small approximation gap observed in [10] suggests that it is plausible that posterior regularization would encourage the true posteriors to be approximately Gaussian.

Given that $\sigma$ modulates the smoothness of the inference model, it is natural to suspect that a larger choice of $\sigma$ results in a stronger regularization. To formalize this notion of regularization strength, we introduce a way to partially order a set of regularizers $\{R_i(\theta)\}$.

**Definition 1.** *Suppose two regularizers $R_1(\theta)$ and $R_2(\theta)$ share the same minimum $\min_\theta R_1(\theta) = \min_\theta R_2(\theta)$. We say that $R_1$ is a stronger regularizer than $R_2$ if $R_1(\theta) \geq R_2(\theta)$ for all $\theta \in \Theta$.*

Note that any two regularizers can be modified via scalar addition to share the same minimum. Furthermore, if $R_1$ is stronger than $R_2$, then $R_1$ and $R_2$ share at least one minimizer. We now apply Definition 1 to characterize the regularization strength of $R(\theta\,;\sigma)$ as $\sigma$ increases.

**Definition 2.** *We say that $\mathcal{F}$ is closed under input translation if $f \in \mathcal{F} \implies f_a \in \mathcal{F}$ for all $a \in \mathcal{X}$, where $f_a(x) = f(x + a)$.*

**Proposition 1.** *Consider the denoising regularizer $R(\theta \,;\sigma)$. Suppose $\mathcal{F}$ is closed under input translation and that, for any $\theta \in \Theta$, there exists $f \in \mathcal{F}$ such that $f(x)$ maps to the prior $p_\theta(z)$ all $x \in \mathcal{X}$. Furthermore, assume that there exists $\theta \in \Theta$ such that $p_\theta(x,z) = p_\theta(z)p_\theta(x)$. Then $R(\theta \,;\sigma_1)$ is stronger $R(\theta \,;\sigma_2)$ when $\sigma_1 \geq \sigma_2$; i.e., $\min_\theta R(\theta \,;\sigma_1) = \min_\theta R(\theta \,;\sigma_2) = 0$ and $R(\theta \,;\sigma_1) \geq R(\theta \,;\sigma_2)$ for all $\theta \in \Theta$.*

Lemma 1 and Proposition 1 show that as we increase $\sigma$, the optimal inference model is forced to become smoother and the regularization strength increases. Figure 1 is consistent with this analysis, showing the progression from under-regularized to over-regularized models as we increase $\sigma$.

It is worth noting that, in addition to adjusting the denoising regularizer strength via $\sigma$, it is also possible to adjust the strength by taking a convex combination of the VAE and DVAE objectives. In particular, we can define the *partially* denoising regularizer $R(\theta \,;\sigma,\alpha)$ as

$$\min_{f \in \mathcal{F}(\mathcal{Q})} \mathbb{E}_{\hat{p}(x)} \left( \alpha \cdot \mathbb{E}_\varepsilon \left[ D(f(x+\varepsilon) \,\|\, p_\theta(z\mid x)) \right] + (1-\alpha) \cdot D(f(x) \,\|\, p_\theta(z\mid x)) \right) \qquad (16)$$

Importantly, we note that $R(\theta \,;\sigma,\alpha)$ is still strictly non-negative and, when combined with the log-likelihood term, still yields a tractable variational lower bound.

### 3.2 Weight-Normalized Amortized Inference

In addition to DVAE, we propose an alternative method that directly restricts $\mathcal{F}$ to the set of smooth functions. To do so, we consider the case where the inference model is a neural network encoder parameterized by weight matrices $\{W_i\}$ and leverage [18]'s weight normalization technique, which proposes to reparameterize the columns $w_i$ of each weight matrix $W$ as

$$w_i = \frac{v_i}{\|v_i\|} \cdot s_i, \qquad (17)$$

where $v_i \in \mathbb{R}^d, s_i \in \mathbb{R}$ are trainable parameters. Since it is possible to modulate the smoothness of the encoder by capping the magnitude of $s_i$, we introduce a new parameter $u_i \in \mathbb{R}$ and define

$$s_i = \min \left\{ \|v_i\|, \left( \frac{H}{1 + \exp(-u_i)} \right) \right\}. \qquad (18)$$

The norm $\|w_i\|$ is thus bounded by the hyperparameter $H$. We denote the weight-normalized regularizer as $R(\theta \,;\mathcal{F}_H)$, where $\mathcal{F}_H$ is the amortization family induced by a $H$-weight-normalized encoder. Under similar assumptions as Proposition 1, it is easy to see that $\min_\theta R(\theta \,;\mathcal{F}_H) = 0$ for any $H \geq 0$ and that $R(\theta \,;\mathcal{F}_{H_1}) \geq R(\theta \,;\mathcal{F}_{H_2})$ for all $\theta \in \Theta$ when $H_1 \leq H_2$ (since $\mathcal{F}_{H_1} \subseteq \mathcal{F}_{H_2}$). We refer to the resulting model as the weight-normalized inference VAE (WNI-VAE) and show in Table 1 that weight-normalized amortized inference can achieve similar performance as DVAE.

### 3.3 Experiments

We conducted experiments on statically binarized MNIST, statically binarized OMNIGLOT, and the Caltech 101 Silhouettes datasets. These datasets have a relatively small amount of training data and are thus susceptible to model overfitting. For each dataset, we used the same decoder architecture across all four models (VAE, DVAE ($\alpha = 0.5$), DVAE ($\alpha = 1.0$), WNI-VAE) and only modified the encoder, and trained all models using Adam [19] (see Appendix E for more details). To approximate the log-likelihood, we proposed to use importance-weighted stochastic variational inference (IW-SVI), an extension of SVI [20] which we describe in detail in Appendix C. Hyperparameter tuning of DVAE's $\sigma$ and WNI-VAE's $\mathcal{F}_H$ is described in Table 7.

Table 1 shows the performance of VAE, DVAE, and WNI-VAE. Regularizing the inference model consistently improved the test set log-likelihood performance. On the MNIST and Caltech 101 Silhouettes datasets, the results also show a consistent reduction of the test set inference gap when the inference model is regularized. We observed differences in the performance of DVAE versus WNI-VAE on the Caltech 101 Silhouettes dataset, suggesting a difference in how denoising and weight normalization regularizes the inference model; an interesting consideration would thus be to combine DVAE and WNI. As a whole, Table 1 demonstrates that AIR benefits the generative model.

The denoising and weight normalization regularizers have respective hyperparameters $\sigma$ and $H$ that control the regularization strength. In Figure 1, we performed an ablation analysis of how adjusting

Table 1: Test set evaluation of VAE, DVAE, and WNI-VAE. The performance metrics are log-likelihood $\ln p_\theta(x)$, the amortized ELBO $\mathcal{L}(x)$, and the inference gap $\Delta_{\text{inf}} = \ln p_\theta(x) - \mathcal{L}(x)$. All three proposed models out-perform VAE across most metrics.

| | MNIST | | | OMNIGLOT | | | CALTECH | | |
| | $-\ln p_\theta(x)$ | $\Delta_{\text{inf}}$ | $-\mathcal{L}(x)$ | $-\ln p_\theta(x)$ | $\Delta_{\text{inf}}$ | $-\mathcal{L}(x)$ | $-\ln p_\theta(x)$ | $\Delta_{\text{inf}}$ | $-\mathcal{L}(x)$ |
|---|---|---|---|---|---|---|---|---|---|
| VAE | $86.93_{\pm 0.04}$ | $8.54_{\pm 0.14}$ | $95.48_{\pm 0.07}$ | $110.32_{\pm 0.16}$ | $12.03_{\pm 0.25}$ | $122.35_{\pm 0.33}$ | $109.14_{\pm 0.28}$ | $28.90_{\pm 0.42}$ | $138.05_{\pm 0.15}$ |
| DVAE ($\alpha = 0.5$) | $86.46_{\pm 0.02}$ | $\mathbf{6.34}_{\pm 0.05}$ | $\mathbf{92.80}_{\pm 0.07}$ | $109.31_{\pm 0.19}$ | $12.56_{\pm 0.18}$ | $121.87_{\pm 0.37}$ | $\mathbf{108.64}_{\pm 0.19}$ | $23.40_{\pm 0.19}$ | $\mathbf{132.04}_{\pm 0.37}$ |
| DVAE ($\alpha = 1.0$) | $86.51_{\pm 0.02}$ | $6.83_{\pm 0.04}$ | $93.35_{\pm 0.06}$ | $110.12_{\pm 0.18}$ | $12.44_{\pm 0.16}$ | $122.56_{\pm 0.34}$ | $108.66_{\pm 0.23}$ | $23.94_{\pm 0.15}$ | $132.60_{\pm 0.15}$ |
| WNI-VAE | $\mathbf{86.42}_{\pm 0.01}$ | $6.68_{\pm 0.01}$ | $93.10_{\pm 0.02}$ | $\mathbf{109.16}_{\pm 0.12}$ | $\mathbf{11.39}_{\pm 0.10}$ | $\mathbf{120.55}_{\pm 0.20}$ | $108.94_{\pm 0.31}$ | $28.88_{\pm 0.29}$ | $137.82_{\pm 0.25}$ |

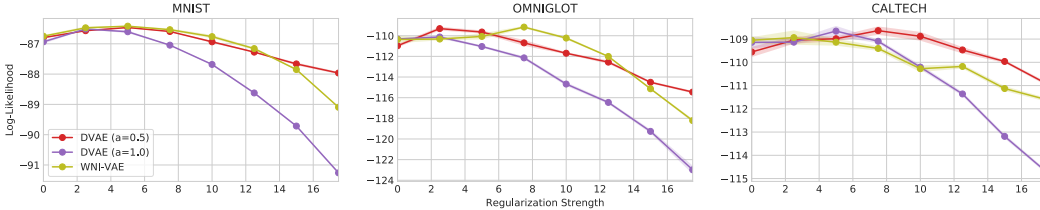

Figure 1: Evaluation of the log-likelihood performance of all three proposed models as we vary the regularization parameter value. The regularization parameter is defined in Table 7. When the parameter value is too small, the model overfits and the test set performance degrades. When the parameter value is too high, the model underfits.

the regularization strength impacts the test set log-likelihood. In almost all cases, we see a transition from overfitting to underfitting as we adjust the strength of AIR. For well-chosen regularization strength, however, it is possible to increase the test set log-likelihood performance by $0.5 \sim 1.0$ nats—a non-trivial improvement.

### 3.4 How Does Amortized Inference Regularization Affect the Generator?

Table 1 shows that regularizing the inference model empirically benefits the generative model. We now provide some initial theoretical characterization of how a smoothed amortized inference model affects the generative model. Our analysis rests on the following proposition.

**Proposition 2.** *Let $\mathcal{P}$ be an exponential family with corresponding mean parameter space $\mathcal{M}$ and sufficient statistic function $T(\cdot)$. With a slight abuse of notation, consider $g \in \mathcal{G} : \mathcal{Z} \to \mathcal{M}$. Define $q(x, z) = \hat{p}(x)q(z \mid x)$, where $q(z \mid x)$ is a fixed inference model. Supposing $\mathcal{G}$ has infinite capacity, then the optimal generative model in Eq. (5) returns $g^*(z) = \mu \in \mathcal{M}$, where*

$$\mu = \sum_{i=1}^{n} q(x^{(i)} \mid z) \cdot T(x^{(i)}) = \sum_{i=1}^{n} \left( \frac{q(z \mid x^{(i)})}{\sum_j q(z \mid x^{(j)})} \cdot T(x^{(i)}) \right). \tag{19}$$

Proposition 2 generalizes the analysis in [21] which determined the optimal generative model when $\mathcal{P}$ is Gaussian. The key observation is that the optimal generative model outputs a convex combination of $\{\phi(x^{(i)})\}$, weighted by $q(x^{(i)} \mid z)$. Furthermore, the weights $q(x^{(i)} \mid z)$ are simply density ratios of the proposal distributions $\{q(z \mid x^{(i)})\}$. As we increase the smoothness of the amortized inference model, the weight $q(x^{(i)} \mid z)$ should tend toward $\frac{1}{n}$ for all $z \in \mathcal{Z}$. This suggests that a smoothed inference model provides a natural way to smooth (and thus regularize) the generative model.

## 4 Amortized Inference Regularization in Importance-Weighted Autoencoders

In this section, we extend AIR to importance-weighted autoencoders (IWAE-$k$). Although the application is straightforward, we demonstrate a noteworthy relationship between the number of importance samples $k$ and the effect of AIR. To begin our analysis, we consider the IWAE-$k$ objective

$$\max_{\theta, \phi} \mathbb{E}_{z_1 \dots z_k \sim q_\phi(z|x)} \left[ \ln \frac{1}{k} \sum_{i=1}^{k} \frac{p_\theta(x, z_i)}{q_\phi(z_i \mid x)} \right], \tag{20}$$

where $\{z_1 \ldots z_k\}$ are $k$ samples from the proposal distribution $q_\phi(z \mid x)$ to be used as importance-samples. Analysis by [22] allows us to rewrite it as a regularized maximum likelihood objective

$$\max_\theta \mathbb{E}_{\hat{p}(x)}[\ln p_\theta(x)] - \overbrace{\min_{f \in \mathcal{F}(\mathcal{Q})} \mathbb{E}_{\hat{p}(x)}\mathbb{E}_{z_2 \ldots z_k \sim f(x)}\tilde{D}(\tilde{f}_k(x, z_1 \ldots z_k) \parallel p_\theta(z \mid x))}^{R_k(\theta)}, \qquad (21)$$

where $\tilde{f}_k$ (or equivalently $\tilde{q}_k$) is the unnormalized distribution

$$\tilde{f}_k(x, z_2 \ldots z_k)(z_1) = \frac{p_\theta(x, z_1)}{\frac{1}{k}\sum_i \frac{p_\theta(x,z_i)}{f(x)(z_i)}} = \tilde{q}_k(z_1 \mid x, z_2 \ldots z_k) \qquad (22)$$

and $\tilde{D}(q \parallel p) = \int q(z)[\ln q(z) - \ln p(z)]\, dz$ is the Kullback-Leibler divergence extended to unnormalized distributions. For notational simplicity, we omit the dependency of $\tilde{f}_k$ on $(z_2 \ldots z_k)$. Importantly, [22] showed that the IWAE with $k$ importance samples drawn from the amortized inference model $f$ is, on expectation, equivalent to a VAE with $1$ importance sample drawn from the more expressive inference model $\tilde{f}_k$.

## 4.1 Importance Sampling Attenuates Amortized Inference Regularization

We now consider the interaction between importance sampling and AIR. We introduce the regularizer $R_k(\theta\,;\sigma, \mathcal{F}_H)$ as follows

$$R_k(\theta\,;\sigma, \mathcal{F}_H) = \min_{f \in \mathcal{F}_H(\mathcal{Q})} \mathbb{E}_{\hat{p}(x)}\mathbb{E}_\varepsilon \mathbb{E}_{z_2 \ldots z_k \sim f(x+\varepsilon)}\tilde{D}(\tilde{f}_k(x + \varepsilon) \parallel p_\theta(z \mid x)), \qquad (23)$$

which corresponds to a regularizer where weight normalization, denoising, and importance sampling are simultaneously applied. By adapting Theorem 1 from [8], we can show that

**Proposition 3.** *Consider the regularizer $R_k(\theta\,;\sigma, \mathcal{F}_H)$. Under similar assumptions as Proposition 1, then $R_{k_1}$ is stronger than $R_{k_2}$ when $k_1 \leq k_2$; i.e., $\min_\theta R_{k_1}(\theta\,;\sigma, \mathcal{F}_H) = \min_\theta R_{k_2}(\theta\,;\sigma, \mathcal{F}_H) = 0$ and $R_{k_1}(\theta\,;\sigma, \mathcal{F}_H) \leq R_{k_2}(\theta\,;\sigma, \mathcal{F}_H)$ for all $\theta \in \Theta$.*

A notable consequence of Proposition 3 is that as $k$ increases, AIR exhibits a weaker regularizing effect on the posterior distributions $\{p_\theta(z \mid x^{(i)})\}$. Intuitively, this arises from the phenomenon that although AIR is applied to $f$, the subsequent importance-weighting procedure can still create a flexible $\tilde{f}_k$. Our analysis thus predicts that AIR is less likely to cause *underfitting* of IWAE-$k$'s generative model as $k$ increases, which we demonstrate in Figure 2. In the limit of infinite importance samples, we also predict AIR to have zero regularizing effect since $\tilde{f}_\infty$ (under some assumptions) can always approximate any posterior. However, for practically feasible values of $k$, we show in Tables 2 and 3 that AIR is a highly effective regularizer.

## 4.2 Experiments

Table 2: Test set evaluation of the four models when trained with $8$ importance samples. $\mathcal{L}_8(x)$ denotes the amortized ELBO using $8$ importance samples. $\Delta_{\text{inf}} = \ln p_\theta(x) - \mathcal{L}_8(x)$.

|  | MNIST | | | OMNIGLOT | | | CALTECH | | |
|---|---|---|---|---|---|---|---|---|---|
|  | $-\ln p_\theta(x)$ | $\Delta_{\text{inf}}$ | $-\mathcal{L}_8(x)$ | $-\ln p_\theta(x)$ | $\Delta_{\text{inf}}$ | $-\mathcal{L}_8(x)$ | $-\ln p_\theta(x)$ | $\Delta_{\text{inf}}$ | $-\mathcal{L}_8(x)$ |
| IWAE | $86.21_{\pm 0.01}$ | $6.13_{\pm 0.03}$ | $92.34_{\pm 0.02}$ | $108.18_{\pm 0.24}$ | $8.69_{\pm 0.39}$ | $116.87_{\pm 0.16}$ | $108.65_{\pm 0.11}$ | $21.52_{\pm 0.13}$ | $130.17_{\pm 0.09}$ |
| DIWAE ($\alpha = 0.5$) | $85.78_{\pm 0.02}$ | $4.47_{\pm 0.02}$ | $90.25_{\pm 0.03}$ | $\mathbf{107.01}_{\pm 0.11}$ | $8.64_{\pm 0.07}$ | $\mathbf{115.66}_{\pm 0.17}$ | $\mathbf{107.34}_{\pm 0.17}$ | $17.61_{\pm 0.18}$ | $124.96_{\pm 0.14}$ |
| DIWAE ($\alpha = 1.0$) | $85.78_{\pm 0.03}$ | $\mathbf{4.21}_{\pm 0.03}$ | $\mathbf{90.00}_{\pm 0.06}$ | $107.47_{\pm 0.06}$ | $\mathbf{8.57}_{\pm 0.14}$ | $116.04_{\pm 0.18}$ | $107.54_{\pm 0.11}$ | $\mathbf{17.06}_{\pm 0.35}$ | $\mathbf{124.60}_{\pm 0.29}$ |
| WNI-IWAE | $85.81_{\pm 0.01}$ | $4.33_{\pm 0.03}$ | $90.14_{\pm 0.04}$ | $107.15_{\pm 0.08}$ | $8.78_{\pm 0.17}$ | $115.93_{\pm 0.10}$ | $107.98_{\pm 0.19}$ | $22.18_{\pm 0.33}$ | $130.16_{\pm 0.14}$ |

Table 3: Test set evaluation of the four models when trained with $64$ importance samples. $\Delta_{\text{inf}} = \ln p_\theta(x) - \mathcal{L}_{64}(x)$.

|  | MNIST | | | OMNIGLOT | | | CALTECH | | |
|---|---|---|---|---|---|---|---|---|---|
|  | $-\ln p_\theta(x)$ | $\Delta_{\text{inf}}$ | $-\mathcal{L}_{64}(x)$ | $-\ln p_\theta(x)$ | $\Delta_{\text{inf}}$ | $-\mathcal{L}_{64}(x)$ | $-\ln p_\theta(x)$ | $\Delta_{\text{inf}}$ | $-\mathcal{L}_{64}(x)$ |
| IWAE | $86.06_{\pm 0.03}$ | $4.41_{\pm 0.10}$ | $90.48_{\pm 0.07}$ | $107.31_{\pm 0.14}$ | $\mathbf{6.66}_{\pm 0.22}$ | $113.97_{\pm 0.10}$ | $108.89_{\pm 0.35}$ | $16.51_{\pm 0.32}$ | $125.40_{\pm 0.25}$ |
| DIWAE ($\alpha = 0.5$) | $\mathbf{85.55}_{\pm 0.02}$ | $\mathbf{3.01}_{\pm 0.01}$ | $\mathbf{88.56}_{\pm 0.02}$ | $\mathbf{106.02}_{\pm 0.01}$ | $6.98_{\pm 0.06}$ | $113.00_{\pm 0.07}$ | $\mathbf{106.94}_{\pm 0.11}$ | $\mathbf{12.28}_{\pm 0.14}$ | $\mathbf{119.22}_{\pm 0.11}$ |
| DIWAE ($\alpha = 1.0$) | $\mathbf{85.55}_{\pm 0.02}$ | $3.15_{\pm 0.02}$ | $88.70_{\pm 0.04}$ | $106.15_{\pm 0.03}$ | $6.70_{\pm 0.05}$ | $\mathbf{112.85}_{\pm 0.07}$ | $106.96_{\pm 0.11}$ | $12.94_{\pm 0.22}$ | $119.87_{\pm 0.16}$ |
| WNI-IWAE | $85.64_{\pm 0.03}$ | $3.10_{\pm 0.01}$ | $88.74_{\pm 0.03}$ | $106.17_{\pm 0.07}$ | $7.11_{\pm 0.07}$ | $113.28_{\pm 0.13}$ | $108.15_{\pm 0.11}$ | $14.42_{\pm 0.20}$ | $122.57_{\pm 0.10}$ |

Tables 2 and 3 extends the model evaluation to IWAE-8 and IWAE-64. We see that the denoising IWAE (DIWAE) and weight-normalized inference IWAE (WNI-IWAE) consistently out-perform the standard IWAE on test set log-likelihood evaluations. Furthermore, the regularized models frequently reduced the inference gap as well. Our results demonstrate that AIR is a highly effective regularizer even when a large number of importance samples are used.

Our main experimental contribution in this section is the verification that increasing the number of importance samples results in less underfitting when the inference model is over-regularized. In contrast to $k = 1$, where aggressively increasing the regularization strength can cause considerable underfitting, Figure 2 shows that increasing the number of importance samples to $k = 8$ and $k = 64$ makes the models much more robust to mis-specified choices of regularization strength. Interestingly, we also observed that the optimal regularization strength (determined using the validation set) increases with $k$ (see Table 7 for details). The robustness of importance sampling when paired with amortized inference regularization makes AIR an effective and practical way to regularize IWAE.

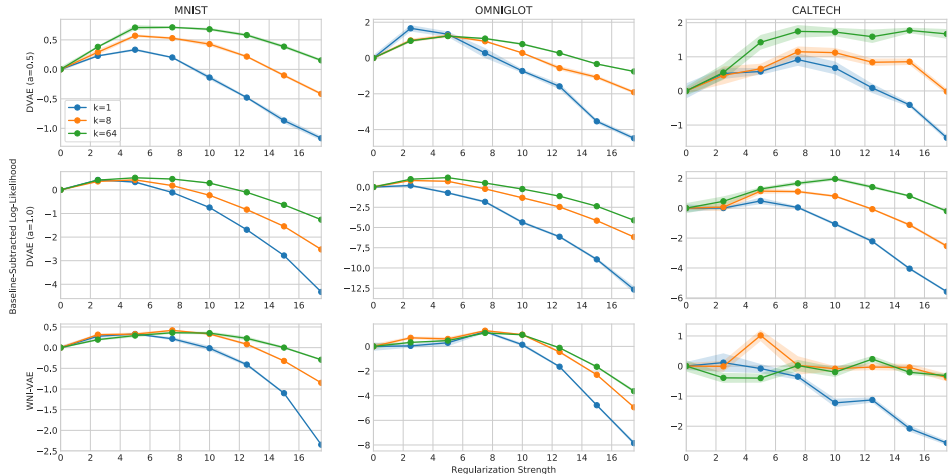

Figure 2: Evaluation of the log-likelihood performance of all three proposed models as we vary the regularization parameter (see Table 7 for definition) and number of importance samples $k$. To compare across different $k$'s, the performance without regularization (IWAE-$k$ baseline) is subtracted. We see that IWAE-64 is the least likely to underfit when the regularization parameter value is high.

### 4.3 Are High Signal-to-Noise Ratio Gradients Necessarily Better?

We note the existence of a related work [23] that also concluded that approximating maximum likelihood training is not necessarily better. However, [23] focused on increasing the signal-to-noise ratio of the gradient updates and analyzed the trade-off between importance sampling and Monte Carlo sampling under budgetary constraints. An in-depth discussion of these two works within the context of generalization is provided in Appendix D.

## 5 Conclusion

In this paper, we challenged the conventional role that amortized inference plays in training deep generative models. In addition to expediting variational inference, amortized inference introduces new ways to regularize maximum likelihood training. We considered a special case of amortized inference regularization (AIR) where the inference model must learn a smoothed mapping from $\mathcal{X} \to \mathcal{Q}$ and showed that the denoising variational autoencoder (DVAE) and weight-normalized inference (WNI) are effective instantiations of AIR. Promising directions for future work include replacing denoising with adversarial training [24] and weight normalization with spectral normalization [25]. Furthermore, we demonstrated that AIR plays a crucial role in the regularization of IWAE, and that higher levels of regularization may be necessary due to the attenuating effects of importance sampling on AIR. We believe that variational family expansion by Monte Carlo methods [26] may exhibit the same attenuating effect on AIR and recommend this as an additional research direction.

**Acknowledgements**

This research was supported by TRI, NSF (#1651565, #1522054, #1733686 ), ONR, Sony, and FLI. Toyota Research Institute provided funds to assist the authors with their research but this article solely reflects the opinions and conclusions of its authors and not TRI or any other Toyota entity.

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
