[Supplementary Material]

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

 | $86.21_{\pm0.01}$ | $6.13_{\pm0.03}$ | $92.34_{\pm0.02}$ | $108.18_{\pm0.24}$ | $8.69_{\pm0.39}$ | $116.87_{\pm0.16}$ | $108.65_{\pm0.11}$ | $21.52_{\pm0.13}$ | $130.17_{\pm0.09}$ |
| DIWAE ($\alpha = 0.5$) | $85.78_{\pm0.02}$ | $4.47_{\pm0.02}$ | $90.25_{\pm0.03}$ | $107.01_{\pm0.11}$ | $8.64_{\pm0.07}$ | $115.66_{\pm0.17}$ | $107.34_{\pm0.17}$ | $17.61_{\pm0.18}$ | $124.96_{\pm0.14}$ |
| DIWAE ($\alpha = 1.0$) | $85.78_{\pm0.03}$ | $4.21_{\pm0.03}$ | $90.00_{\pm0.06}$ | $107.47_{\pm0.06}$ | $\mathbf{8.57}_{\pm0.14}$ | $116.04_{\pm0.18}$ | $107.54_{\pm0.11}$ | $17.06_{\pm0.35}$ | $\mathbf{124.60}_{\pm0.29}$ |
| WNI-IWAE | $85.81_{\pm0.01}$ | $4.33_{\pm0.03}$ | $90.14_{\pm0.04}$ | $107.15_{\pm0.08}$ | $8.78_{\pm0.17}$ | $115.93_{\pm0.10}$ | $107.98_{\pm0.19}$ | $22.18_{\pm0.33}$ | $130.16_{\pm0.14}$ |

Table 3: Test set evaluation of the four models when trained with $64$ importance samples. $\Delta_{\text{inf}} = \ln p_\theta(x) - \mathcal{L}_{64}(x)$.

| | MNIST | | | OMNIGLOT | | | CALTECH | | |
|---|---|---|---|---|---|---|---|---|---|
| | $-\ln p_\theta(x)$ | $\Delta_{\text{inf}}$ | $-\mathcal{L}_{64}(x)$ | $-\ln p_\theta(x)$ | $\Delta_{\text{inf}}$ | $-\mathcal{L}_{64}(x)$ | $-\ln p_\theta(x)$ | $\Delta_{\text{inf}}$ | $-\mathcal{L}_{64}(x)$ |
| IWAE | $86.06_{\pm0.03}$ | $4.41_{\pm0.10}$ | $90.48_{\pm0.07}$ | $107.31_{\pm0.14}$ | $\mathbf{6.66}_{\pm0.22}$ | $113.97_{\pm0.10}$ | $108.89_{\pm0.35}$ | $16.51_{\pm0.32}$ | $125.40_{\pm0.25}$ |
| DIWAE ($\alpha = 0.5$) | $85.55_{\pm0.02}$ | $3.01_{\pm0.01}$ | $88.56_{\pm0.02}$ | $106.02_{\pm0.01}$ | $6.98_{\pm0.06}$ | $113.00_{\pm0.07}$ | $106.94_{\pm0.11}$ | $12.28_{\pm0.14}$ | $\mathbf{119.22}_{\pm0.11}$ |
| DIWAE ($\alpha = 1.0$) | $85.55_{\pm0.02}$ | $3.15_{\pm0.02}$ | $88.70_{\pm0.04}$ | $106.15_{\pm0.03}$ | $6.70_{\pm0.05}$ | $\mathbf{112.85}_{\pm0.07}$ | $106.96_{\pm0.11}$ | $12.94_{\pm0.22}$ | $119.87_{\pm0.16}$ |
| WNI-IWAE | $85.64_{\pm0.03}$ | $3.10_{\pm0.01}$ | $88.74_{\pm0.03}$ | $106.17_{\pm0.07}$ | $7.11_{\pm0.07}$ | $113.28_{\pm0.13}$ | $108.15_{\pm0.11}$ | $14.42_{\pm0.20}$ | $122.57_{\pm0.10}$ |

Tables 2 and 3 extends the model evaluation to IWAE-8 and IWAE-64. We see that the denoising IWAE (DIWAE) and weight-normalized inference IWAE (WNI-IWAE) consistently out-perform the standard IWAE on test set log-likelihood evaluations. Furthermore, the regularized models frequently reduced the inference gap as well. Our results demonstrate that AIR is a highly effective regularizer even when a large number of importance samples are used.

Our main experimental contribution in this section is the verification that increasing the number of importance samples results in less underfitting when the inference model is over-regularized. In contrast to $k = 1$, where aggressively increasing the regularization strength can cause considerable underfitting, Figure 2 shows that increasing the number of importance samples to $k = 8$ and $k = 64$ makes the models much more robust to mis-specified choices of regularization strength. Interestingly, we also observed that the optimal regularization strength (determined using the validation set) increases with $k$ (see Table 7 for details). The robustness of importance sampling when paired with amortized inference regularization makes AIR an effective and practical way to regularize IWAE.

Figure 2: Evaluation of the log-likelihood performance of all three proposed models as we vary the regularization parameter (see Table 7 for definition) and number of importance samples $k$. To compare across different $k$'s, the performance without regularization (IWAE-$k$ baseline) is subtracted. We see that IWAE-64 is the least likely to underfit when the regularization parameter value is high.

## 4.3 Are High Signal-to-Noise Ratio Gradients Necessarily Better?

We note the existence of a related work [23] that also concluded that approximating maximum likelihood training is not necessarily better. However, [23] focused on increasing the signal-to-noise ratio of the gradient updates and analyzed the trade-off between importance sampling and Monte Carlo sampling under budgetary constraints. An in-depth discussion of these two works within the context of generalization is provided in Appendix D.

## 5 Conclusion

In this paper, we challenged the conventional role that amortized inference plays in training deep generative models. In addition to expediting variational inference, amortized inference introduces new ways to regularize maximum likelihood training. We considered a special case of amortized inference regularization (AIR) where the inference model must learn a smoothed mapping from $\mathcal{X} \rightarrow \mathcal{Q}$ and showed that the denoising variational autoencoder (DVAE) and weight-normalized inference (WNI) are effective instantiations of AIR. Promising directions for future work include replacing denoising with adversarial training [24] and weight normalization with spectral normalization [25]. Furthermore, we demonstrated that AIR plays a crucial role in the regularization of IWAE, and that higher levels of regularization may be necessary due to the attenuating effects of importance sampling on AIR. We believe that variational family expansion by Monte Carlo methods [26] may exhibit the same attenuating effect on AIR and recommend this as an additional research direction.

**Acknowledgements**

This research was supported by TRI, NSF (#1651565, #1522054, #1733686 ), ONR, Sony, and FLI. Toyota Research Institute provided funds to assist the authors with their research but this article solely reflects the opinions and conclusions of its authors and not TRI or any other Toyota entity.

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

## A  Overly Expressive Amortization Family Hurts Generalization

In the experiments by [10], they observed that an overly expressive amortization family increases the test set inference gap, but does not impact the test set log-likelihood. We show in Table 4 that [10]'s observation is not true in general, and that an overly expressive amortization family can in fact hurt test set log-likelihood. Details regarding the architectures are provided in Appendix E.

Table 4: Performance evaluation when an over-expressive amortization family is used (i.e. a larger encoder). Comparison is made against models that use a smaller encoder. The results show that using a large encoder consistently hurts generalization by over 1 nat.

| | MNIST ($k=1$) | | | MNIST ($k=8$) | | | MNIST ($k=64$) | | |
|---|---|---|---|---|---|---|---|---|---|
| | $-\ln p_\theta(x)$ | $\Delta_{\text{inf}}$ | $-\mathcal{L}_1(x)$ | $-\ln p_\theta(x)$ | $\Delta_{\text{inf}}$ | $-\mathcal{L}_8(x)$ | $-\ln p_\theta(x)$ | $\Delta_{\text{inf}}$ | $-\mathcal{L}_6 4(x)$ |
| IWAE (Large Encoder) | $87.43_{\pm 0.05}$ | $11.32_{\pm 0.21}$ | $98.74_{\pm 0.25}$ | $86.98_{\pm 0.07}$ | $8.00_{\pm 0.18}$ | $94.98_{\pm 0.17}$ | $86.70_{\pm 0.06}$ | $5.91_{\pm 0.11}$ | $92.61_{\pm 0.10}$ |
| IWAE | $86.93_{\pm 0.04}$ | $8.54_{\pm 0.14}$ | $95.48_{\pm 0.07}$ | $86.21_{\pm 0.01}$ | $6.13_{\pm 0.03}$ | $92.34_{\pm 0.02}$ | $86.06_{\pm 0.03}$ | $4.41_{\pm 0.10}$ | $90.48_{\pm 0.07}$ |
| DIWAE ($\alpha = 0.5$) | $86.46_{\pm 0.02}$ | $\mathbf{6.34}_{\pm 0.05}$ | $\mathbf{92.80}_{\pm 0.07}$ | $\mathbf{85.78}_{\pm 0.02}$ | $4.47_{\pm 0.02}$ | $90.25_{\pm 0.03}$ | $\mathbf{85.55}_{\pm 0.02}$ | $\mathbf{3.01}_{\pm 0.01}$ | $\mathbf{88.56}_{\pm 0.02}$ |
| DIWAE ($\alpha = 1.0$) | $86.51_{\pm 0.02}$ | $6.83_{\pm 0.04}$ | $93.35_{\pm 0.06}$ | $\mathbf{85.78}_{\pm 0.03}$ | $\mathbf{4.21}_{\pm 0.03}$ | $\mathbf{90.00}_{\pm 0.06}$ | $\mathbf{85.55}_{\pm 0.02}$ | $3.15_{\pm 0.02}$ | $88.70_{\pm 0.04}$ |
| WNI-IWAE | $\mathbf{86.42}_{\pm 0.01}$ | $6.68_{\pm 0.01}$ | $93.10_{\pm 0.02}$ | $85.81_{\pm 0.01}$ | $4.33_{\pm 0.03}$ | $90.14_{\pm 0.04}$ | $85.64_{\pm 0.03}$ | $3.10_{\pm 0.01}$ | $88.74_{\pm 0.03}$ |

## B  Revisiting [17]'s Denoising Variational Autoencoder Analysis

In [17]'s Lemma 1, they considered a joint distribution $p_\theta(x, z)$. They introduced an auxiliary variable $z'$ into their inference model (here $z'$ takes on the role of the perturbed input $\tilde{x} = x + \varepsilon$. To avoid confusion, we stick to the notation used in their Lemma) and considered the inference model

$$q_\varphi(z \mid z')q_\psi(z' \mid x). \tag{24}$$

They considered two ways to use this inference model. The first approach is to marginalize the auxiliary latent variable $z'$. This defines the resulting inference model

$$q_\phi(z \mid x) = \int q_\varphi(z \mid z')q_\psi(z' \mid x)dz'. \tag{25}$$

This yields the lower bound

$$\mathcal{L}_a = \mathbb{E}_{q_\phi(z|x)}\left[\ln \frac{p_\theta(x, z)}{q_\phi(z \mid x)}\right]. \tag{26}$$

Next, they considered an alternative lower bound

$$\mathcal{L}_b = \mathbb{E}_{q_\varphi(z|z')q_\psi(z'|x)}\left[\ln \frac{p_\theta(x, z)}{q_\varphi(z \mid z')}\right]. \tag{27}$$

[17]'s Lemma 1 claims that

1. $\mathcal{L}_a$ and $\mathcal{L}_b$ are valid lower bounds of $\ln p_\theta(x)$
2. $\mathcal{L}_b \geq \mathcal{L}_a$.

Using Lemma 1, [17] motivated the denoising variational autoencoder by concluding that it provides a tighter bound than marginalization of the noise variable. Although statement 1 is correct, statement 2 is not. Their proof of statement 2 is presented as follows

$$\mathbb{E}_{q_\varphi(z|z')q_\psi(z'|x)}\left[\ln \frac{p_\theta(x, z)}{q_\varphi(z \mid z')}\right] \stackrel{?}{=} \mathbb{E}_{q_\phi(z|x)}\left[\ln \frac{p_\theta(x, z)}{q_\varphi(z \mid z')}\right] \tag{28}$$

$$= \mathbb{E}_{q_\phi(z|x)}\left[\log p_\theta(x, z)\right] - \mathbb{E}_{q_\phi(z|x)}\left[\ln q_\phi(z \mid z')\right] \tag{29}$$

$$\geq \mathbb{E}_{q_\phi(z|x)}\left[\log p_\theta(x, z)\right] - \mathbb{E}_{q_\phi(z|x)}\left[\ln q_\phi(z \mid z')\right] \tag{30}$$

$$= \mathbb{E}_{q_\phi(z|x)}\left[\ln \frac{p_\theta(x, z)}{q_\phi(z \mid x)}\right] \tag{31}$$

We indicate the mistake with $\stackrel{?}{=}$; their proof of statement 2 relied on the assumption that

$$\mathbb{E}_{q_\varphi(z|z')q_\psi(z'|x)}\left[\ln \frac{p_\theta(x, z)}{q_\varphi(z \mid z')}\right] = \mathbb{E}_{q_\phi(z|x)}\left[\ln \frac{p_\theta(x, z)}{q_\varphi(z \mid z')}\right]. \tag{32}$$

Crucially, the RHS is ill-defined since it does not take the expectation over $z'$, whereas the LHS explicitly specifies an expectation over $z' \sim q_\psi(z' \mid x)$. This difference, while subtle, invalidates the subsequent steps. If we fix Eq. (28) and attempt to see if the rest of the proof still follows, we will find that

$$\mathbb{E}_{q_\varphi(z|z')q_\psi(z'|x)}\left[\ln \frac{p_\theta(x,z)}{q_\varphi(z \mid z')}\right] = \mathbb{E}_{q_\phi(z|x)}\left[\log p_\theta(x,z)\right] - \mathbb{E}_{q_\varphi(z|z')q_\psi(z'|x)}\left[\ln q_\psi(z \mid z')\right] \quad (33)$$

$$\not\geq \mathbb{E}_{q_\phi(z|x)}\left[\log p_\theta(x,z)\right] - \mathbb{E}_{q_\varphi(z|z')q_\psi(z'|x)}\left[\ln q_\phi(z \mid x)\right] \quad (34)$$

$$= \mathbb{E}_{q_\phi(z|x)}\left[\ln \frac{p_\theta(x,z)}{q_\phi(z \mid x)}\right]. \quad (35)$$

Indeed, the inequality will point the other way since

$$\mathbb{E}_{q_\varphi(z|z')q_\psi(z'|x)}\left[\ln q_\psi(z \mid z') - \ln q_\phi(z \mid x)\right] = \mathbb{E}_{q_\psi(z'|x)}\mathbb{E}_{q_\varphi(z|z')}\ln \frac{q_\varphi(z \mid z')}{q_\phi(z \mid x)} \quad (36)$$

$$= \mathbb{E}_{q_\psi(z'|x)}D(q_\varphi(z \mid z') \parallel q_\phi(z \mid x)) \quad (37)$$

$$\geq 0 \implies \quad (38)$$

$$-\mathbb{E}_{q_\varphi(z|z')q_\psi(z'|x)}\left[\ln q_\psi(z \mid z')\right] \leq -\mathbb{E}_{q_\varphi(z|z')q_\psi(z'|x)}\left[\ln q_\phi(z \mid x)\right]. \quad (39)$$

Their conclusion that marginalizing over the noise variable results in a looser bound is thus incorrect. In the text (beneath [17] Eq. (11)), they further implied that the denoising VAE and standard VAE objectives are not comparable. We show in Proposition 1 that the denoising VAE objective is in fact a lower bound of the standard VAE objective.

## C   Importance-Weighted Stochastic Variational Inference

(a)                              (b)                              (c)

Figure 3: Evaluation of IW-SVI versus IWAE-$k$ for a fixed generative model. IW-SVI out-performss IWAE-$k$ on both computation time and number of importance samples needed. Similar to [11], we conclude that IWAE-$k$'s poor approximation of the log-likelihood is attributable to an overfit amortized inference model. Fig. 3a) IW-SVI computation time depends on the number of gradient update steps. IWAE-$k$ computation time depends on the number of importance samples $k$. IWAE-100000 still under-performs IW-SVI ($k = 5000, \ell = 1, T = 100$), demonstrating the efficacy of IW-SVI. Fig. 3b) Comparison of IWAE and IW-SVI ($T = 3000$) for different values of $k$. Fig. 3c Comparison of IW-SVI ($k = 5000$) for different values of $T$.

We propose a simple method to approximate the marginal $\ln p_\theta(x)$. A common approach for approximating the log marginal is the IWAE-5000 [7, 8, 27, 28], which proposes to compute $\mathcal{L}_{5000}(x\,; \theta, \phi)$ where

$$\ln p_\theta(x) \geq \mathcal{L}_k(x\,; \theta, \phi) = \mathbb{E}_{z_1\ldots z_k \sim q_\phi(z|x)}\left(\ln \frac{1}{k}\sum_{i=1}^{k}\frac{p_\theta(x,z^{(i)})}{q_\phi(z^{(i)} \mid x)}\right). \quad (40)$$

However, this approach relies on the learned inference model $q_\phi(z \mid x)$, which might overfit to the training set. To address this issue, we propose to perform importance-weighted stochastic variational

inference (IW-SVI)

$$\ln p_\theta(x) \geq \mathcal{L}_k(x\,;\theta, q_{x,\ell}^*) = \mathbb{E}_{z_1\ldots z_k \sim q_{x,\ell}^*(z)} \left( \ln \frac{1}{k} \sum_{i=1}^{k} \frac{p_\theta(z\mid x)}{q_{x,\ell}^*(z)} \right), \tag{41}$$

$$\text{where } q_{x,\ell}^* = \operatorname*{arg\,max}_{q\in\mathcal{Q}} \mathcal{L}_\ell(x\,;\theta, q). \tag{42}$$

The optimization in Eq. (42) is approximate with $T$ gradient steps. As $k$ and $\ell$ increase, the approximation will approach the true log-likelihood. We approximate log-likelihood over the entire test set using $\mathbb{E}_{\hat{p}_{\text{test}}(x)}\mathcal{L}_k(p_\theta, q_{x,\ell}^*\,;x)$. To reduce speed and memory cost during the per-sample optimization in Eq. (42), we use a large $k = 5000$ but smaller $\ell = 8$, and approximately solved the optimization problem using $T = 3000$ gradient steps. In comparison to IWAE-5000, we consistently observe significant improvement in the log-likelihood approximation. IW-SVI provides a simple alternative to Annealed Importance Sampling, requiring minimal modification to any existing IWAE-$k$ implementation.

## D   Are High Signal-to-Noise Ratio Gradients Necessarily Better?

Our paper shares a similar high level message with a recent study by [23]: that approximating maximum likelihood training is not necessarily better. However, we approach this message in very different ways. [23] observed that importance sampling weakens the signal-to-noise ratio of the gradients used to update the amortized inference model. In response, they proposed to increase this ratio by increasing the number of Monte Carlo samples $m$ used to estimate the expectation in Eq. (5). Under a fixed budget of $T \geq mk$ (where $k$ is the number of importance samples and $m$ is the number of Monte Carlo samples), they observed that it may be desirable to trade off $k$ in order to increase $m$. Given an infinite budget, however, [23]'s hypothesis would still conclude to increase $k$ as much as possible in order to approximate maximum likelihood training.

In contrast, we argue that it may be inherently desirable to regularize the maximum likelihood objective, and that amortized inference regularization is an effective means of doing so. From the perspective of generalization, it is also worth wondering whether high signal-to-noise ratio gradients are necessarily better. The desirability of noisy gradients for improving generalization is an active area of research [29, 30, 31, 32], and an extensive investigation of the role of gradient stochasticity in regularizing the amortized inference model is beyond the scope of our paper. To encourage future exploration in this direction, we show in Figure 4 that the effect of gradient stochasticity is non-negligible. For the standard VAE, we observed that increasing $m$ can cause the model to overfit (on the amortized ELBO objective) over the course of training. Interestingly, we observed that DVAE does not experience this overfitting effect, suggesting that AIR is robust to larger values of $m$.

Figure 4: Comparison of the test set amortized ELBO during training for VAE and DVAE as we vary the number of importance samples $k$ and the number of Monte Carlo samples $m$. In contrast to DVAE, VAE is susceptible to overfitting when $m$ is increased.

# E Experimental Details

**Datasets.** We carried out experiments on the static MNIST, static OMNIGLOT, and Caltech 101 Silhouettes datasets. OMNIGLOT was statically binarized at the beginning of training via random sampling using the pixel real-values as Bernoulli parameters. Training, validation, and test split sizes are provided in Table 5. The MNIST validation set was created by randomly holding out $10000$ samples from the original $60000$-sample training set. The OMNIGLOT validation set was similarly created by randomly holding out $1345$ samples from the original $24345$-sample training set.

Table 5: Training, validation and test splits for each dataset.

| Dataset | Training Split | Validation Split | Test Split |
|---|---|---|---|
| MNIST | 50000 | 10000 | 10000 |
| OMNIGLOT | 23000 | 1345 | 8070 |
| CALTECH | 4100 | 2264 | 2307 |

**Training parameters.** Important training parameters are provided in Table 6. We used the Adam optimizer and exponentially decayed the initial learning rate according to the formula

$$\alpha_t = \alpha_0 \cdot (0.1)^{\frac{t}{T-1}}, \tag{43}$$

where $t \in \{0, \ldots, T-1\}$ is the current iteration and $T$ is the total number of iterations. Early-stopping is applied according to IWAE-5000 evaluation on the validation set.

Table 6: Training parameters used for each dataset. The same architecture is used for all models, with minor modification for WNI-VAE (to account for the weight-normalization implementation). In all cases, we use a Bernoulli decoder and a Gaussian encoder. Notation: d300 denotes a dense layer with ELU activation and 300 output units. z64 denotes 1) a dense layer with $64$ output units (represents the mean of $z$) and 2) a dense layer with softplus activation and $64$ output units (represents the variance of $z$). x784 denotes a dense layer with $784$ output units (represents the logits for $x$)

| Dataset | Encoder Architecture | Decoder Architecture | Initial Learning Rate | Training Iterations | Batch Size |
|---|---|---|---|---|---|
| MNIST (Appendix A) | d1000-d1000-d1000-z64 | d300-d300-x784 | $10^{-3}$ | $1.5 \times 10^6$ | 100 |
| MNIST | d300-d300-z64 | d300-d300-x784 | $10^{-3}$ | $1.5 \times 10^6$ | 100 |
| OMNIGLOT | d200-d200-z64 | d200-d200-x784 | $10^{-3}$ | $1.5 \times 10^6$ | 100 |
| CALTECH | d500-z64 | d500-x784 | $10^{-4}$ | $4 \times 10^5$ | 10 |

**Regularization strength tuning.** The denoising and weight normalization regularizers have hyper-parameters $\sigma$ and $H$ respectively. See Table 7 for hyperparameter search space details. We performed a basic grid search and tuned the regularization strength hyperparameters based on the validation set.

Table 7: The regularization parameter is chosen applied based on hyperparameter tuning on the validation set. Rather than selecting for $\sigma$ or $H$ directly, we reparameterized the search space as $\sigma \cdot \sqrt{d}$ and $\frac{10}{H}$, where $d$ denotes the sample dimensionality, i.e., $\mathcal{X} = \mathbb{R}^d$. Coincidentally, we found that this reparameterization allowed us to use the same search space for both DIWAE and WNI-IWAE. We introduce the convention that setting $\frac{10}{H}$ to zero indicates setting $H = +\infty$. Via this convention, setting $\sigma \cdot \sqrt{d} = \frac{10}{H} = 0$ corresponds to the standard VAE. We restricted the search space to the set $\{2.5, 5.0, \ldots, 17.5\}$, deliberately omitting $\{0.0\}$ to not encompass the baseline (standard VAE).

| | | MNIST | | OMNIGLOT | | CALTECH | |
|---|---|---|---|---|---|---|---|
| | $k$ | $\sigma \cdot \sqrt{d}$ | $\frac{10}{H}$ | $\sigma \cdot \sqrt{d}$ | $\frac{10}{H}$ | $\sigma \cdot \sqrt{d}$ | $\frac{10}{H}$ |
| DIWAE ($\alpha = 0.5$) | 1 | 5.0 | - | 2.5 | - | 7.5 | - |
| | 8 | 5.0 | - | 5.0 | - | 7.5 | - |
| | 64 | 7.5 | - | 5.0 | - | 15.0 | - |
| DIWAE ($\alpha = 1.0$) | 1 | 2.5 | - | 2.5 | - | 5.0 | - |
| | 8 | 5.0 | - | 5.0 | - | 7.5 | - |
| | 64 | 5.0 | - | 5.0 | - | 10.0 | - |
| WNI-IWAE | 1 | - | 5.0 | - | 7.5 | - | 2.5 |
| | 8 | - | 7.5 | - | 7.5 | - | 5.0 |
| | 64 | - | 10.0 | - | 7.5 | - | 12.5 |

# F Proofs

**Remark.** Some of the proofs mention the notion of an infinite capacity $\mathcal{F}$, $\mathcal{G}$ or $\mathcal{Q}$. To clarify, we say that $\mathcal{F}$ has infinite capacity if it is the set of all possible functions that map from $\mathcal{X}$ to $\mathcal{Q}$. Analogously, $\mathcal{G}$ has infinite capacity if it is the set of all possible functions that map from $\mathcal{Z}$ to $\mathcal{P}$. We say that $\mathcal{Q}$ has infinite capacity if it is the set of all possible distributions over the space $\mathcal{Z}$.

**Lemma 1.** *For fixed $(\theta, \sigma, \mathcal{Q})$ and infinite capacity $\mathcal{F}$, the inference model that optimizes the DVAE objective in Eq.* (13) *is the kernel regression model*

$$f_\sigma^*(x) = \arg\min_{q \in \mathcal{Q}} \sum_{i=1}^{n} w_\sigma(x, x^{(i)}) \cdot D(q(z) \parallel p_\theta(z \mid x^{(i)})), \tag{14}$$

*where $w_\sigma(x, x^{(i)}) = \frac{K_\sigma(x, x^{(i)})}{\sum_j K_\sigma(x, x^{(j)})}$ and $K_\sigma(x, y) = \exp\left(-\frac{\|x-y\|}{2\sigma^2}\right)$ is the RBF kernel.*

*Proof.* Define $\tilde{x} = x + \varepsilon$ and $\hat{p}(x, \tilde{x}) = \hat{p}(x)\mathcal{N}(\tilde{x} \mid x, \sigma\mathbf{I})$. Rewrite the objective as

$$R(\theta \, ; \sigma) = \min_{f \in \mathcal{F}(\mathcal{Q})} \mathbb{E}_{\hat{p}(x,\tilde{x})} \left[ D(f(\tilde{x}) \parallel p_\theta(z \mid x)) \right] \tag{44}$$

$$\geq \mathbb{E}_{\hat{p}(\tilde{x})} \min_{q \in \mathcal{Q}} \mathbb{E}_{\hat{p}(x|\tilde{x})} \left[ D(q(z) \parallel p_\theta(z \mid x)) \right]. \tag{45}$$

Recall that $\mathcal{F}$ has infinite capacity. This lower bound is tight since we can select $f_\sigma^* \in \mathcal{F}$ such that

$$f_\sigma^*(\tilde{x}) = \arg\min_{q \in \mathcal{Q}} \mathbb{E}_{\hat{p}(x|\tilde{x})} D(q(z) \parallel p_\theta(z \mid x)). \tag{46}$$

Reexpressing Eq. (46) by expanding $\hat{p}(x \mid \tilde{x})$ yields Eq. (14). $\qquad\square$

**Theorem 1.** *Let $\mathcal{Q}$ be a minimal exponential family with corresponding natural parameter space $\Omega$. With a slight abuse of notation, consider $f \in \mathcal{F} : \mathcal{X} \to \Omega$. Under the simplifying assumption that $p_\theta(z \mid x^{(i)})$ is contained within $\mathcal{Q}$ and parameterized by $\eta^{(i)} \in \Omega$, and that $\mathcal{F}$ has infinite capacity, then the optimal inference model in Lemma 1 returns $f_\sigma^*(x) = \eta \in \Omega$, where*

$$\eta = \sum_{i=1}^{n} w_\sigma(x, x^{(i)}) \cdot \eta^{(i)} \tag{15}$$

*and Lipschitz constant of $f_\sigma^*$ is bounded by $O(1/\sigma^2)$.*

*Proof.* Proof provided in two parts.

**Part 1.** The Kullback-Leibler divergence can be represented as a Bregman divergence associated with the strictly convex log-partition function $A$ of the minimal exponential family as follows

$$D(\eta \parallel \eta^{(i)}) = d_A(\eta^{(i)}, \eta) = A(\eta^{(i)}) - A(\eta) - \nabla A(\eta)^\top (\eta^{(i)} - \eta). \tag{47}$$

Proposition 1 from [33] shows that that for any convex combination weights $\{w_i\}, \sum_{i=1}^{n} w_i = 1$, the minimizer of a weighted average of Bregman divergences is

$$\sum_{i=1}^{n} w_i x_i = \arg\min_{y \in \Omega} \sum_{i=1}^{n} w_i d_A(x_i, y). \tag{48}$$

It thus follows that

$$f_\sigma^*(x) = \arg\min_{\eta \in \Omega} \sum_{i=1}^{n} w_\sigma(x, x^{(i)}) \cdot D(\eta \parallel \eta^{(i)}) \tag{49}$$

$$= \arg\min_{\eta \in \Omega} \sum_{i=1}^{n} w_\sigma(x, x^{(i)}) \cdot d_A(\eta^{(i)}, \eta) \tag{50}$$

$$= \sum_{i=1}^{n} w_\sigma(x, x^{(i)}) \cdot \eta^{(i)}. \tag{51}$$

**Part 2.** We will write the derivative $\nabla_x f_\sigma^*(x)$ in matrix form by the following notation

$$\nabla_x W_\sigma(x) = \begin{pmatrix} \nabla_x w_\sigma(x, x^{(1)}) & \cdots & \nabla_x w_\sigma(x, x^{(m)}) \end{pmatrix}$$

$$M = \begin{pmatrix} \eta^{(1)} & \cdots & \eta^{(m)} \end{pmatrix}$$

where we also suppose input space $x$ is $n$-dimensional, latent parameter space $\Omega$ is $d$-dimensional, and there are $m$ training examples. Then

$$\nabla_x f_\sigma^*(x) = M \nabla_x W_\sigma(x)^T$$

Let $\|\cdot\|_1$ be the induced 1-norm for matrices, then by the sub-multiplicative property

$$\|\nabla_x f^*(x)\|_1 \leq \|M\|_1 \|\nabla_x W_\sigma(x)^T\|_1$$

Since $\|M\|_1$ is a constant with respect to $\sigma$, we only have to bound $\|\nabla_x W_\sigma(x)^T\|_1$. To do this we study the derivative of $\nabla_x w_\sigma(x, x^{(i)})$, where

$$
\begin{aligned}
\nabla_x w_\sigma(x, x^{(i)}) &= \nabla_x \frac{K_\sigma(x, x^{(i)})}{\sum_j K_\sigma(x, x^{(j)})} \\
&= \frac{K(x, x^{(i)}) \frac{x^{(i)} - x}{\sigma^2} \sum_j K_\sigma(x, x^{(j)}) + K(x, x^{(i)}) \sum_j K(x, x^{(j)}) \frac{x - x^{(j)}}{\sigma^2}}{(\sum_j K_\sigma(x, x^{(j)}))^2} \\
&= \frac{K(x, x^{(i)}) \sum_j K_\sigma(x, x^{(j)}) \frac{x^{(i)} - x^{(j)}}{\sigma^2}}{(\sum_j K_\sigma(x, x^{(j)}))^2}
\end{aligned}
$$

Let $|\cdot|$ denote taking element-wise absolute value, and $x \leq^* y$ denotes for all elements of the vector $|x_i| \leq |y_i|$. By Cauchy inequality and $\|\cdot\|_2 \leq \|\cdot\|_1$ we have

$$\nabla_x w_\sigma(x, x^{(i)}) \leq^* \frac{K(x, x^{(i)}) \sum_j K(x, x^{(j)}) \sum_j |x^{(i)} - x^{(j)}|}{\sigma^2 (\sum_j K_\sigma(x, x^{(j)}))^2} \leq^* \frac{1}{\sigma^2} \sum_j |x^{(i)} - x^{(j)}|$$

Therefore

$$\sup_x \|\nabla_x w_\sigma(x, x^{(i)})\|_1 = O(1/\sigma^2)$$

This gives us a bound on the matrix 1-norm

$$\sup_x \|\nabla_x W_\sigma(x)^T\|_1 \leq \sup_x \sqrt{mn} \|\nabla_x W_\sigma(x)^T\|_\infty = \sqrt{mn} \sup_x \max_{i=1}^n \|\nabla_x w_\sigma(x, x^{(i)})\|_1 = O(1/\sigma^2)$$

Because both $\Omega$ and $\mathcal{X}$ are convex sets, this implies the following Lipschitz property

$$\frac{\|f^*(x_1) - f^*(x_2)\|_1}{\|x_1 - x_2\|_1} \leqslant \sup_x \|\nabla_x f^*(x)\|_1 = O(1/\sigma^2)$$

$\square$

**Proposition 1.** *Consider the denoising regularizer $R(\theta \,;\, \sigma)$. Suppose $\mathcal{F}$ is closed under input translation and that, for any $\theta \in \Theta$, there exists $f \in \mathcal{F}$ such that $f(x)$ maps to the prior $p_\theta(z)$ all $x \in \mathcal{X}$. Furthermore, assume that there exists $\theta \in \Theta$ such that $p_\theta(x, z) = p_\theta(z)p_\theta(x)$. Then $R(\theta \,;\, \sigma_1)$ is stronger $R(\theta \,;\, \sigma_2)$ when $\sigma_1 \geq \sigma_2$; i.e., $\min_\theta R(\theta \,;\, \sigma_1) = \min_\theta R(\theta \,;\, \sigma_2) = 0$ and $R(\theta \,;\, \sigma_1) \geq R(\theta \,;\, \sigma_2)$ for all $\theta \in \Theta$.*

*Proof.* Proof is provided in two parts.

**Part 1.** Recall that $R$ is always non-negative. Since there exists $\theta \in \Theta$ such that $p_\theta(x, z) = p_\theta(z)p_\theta(x)$, and $f \in \mathcal{F}$ such that $f(x) = p_\theta(z)$, then $\min_\theta R(\theta \,;\, \sigma) = 0$ for any choice of $\sigma$.

**Part 2.** Let $\varepsilon_1 \sim \mathcal{N}(\mathbf{0}, \sigma_1 \mathbf{I})$, $\varepsilon_2 \sim \mathcal{N}(\mathbf{0}, \sigma_2 \mathbf{I})$, and $\varepsilon_\delta = \mathcal{N}(\mathbf{0}, (\sigma_1 - \sigma_2)\mathbf{I})$. Then

$$R(\theta \,;\, \sigma_1) = \min_{f \in \mathcal{F}} \mathbb{E}_{\varepsilon_1} \mathbb{E}_{\hat{p}(x)} \left[ D(f(x + \varepsilon_1) \,\|\, p_\theta(z \mid x)) \right] \tag{52}$$

$$= \min_{f \in \mathcal{F}} \mathbb{E}_{\varepsilon_\delta} \mathbb{E}_{\varepsilon_2} \mathbb{E}_{\hat{p}(x)} \left[ D(f(x + \varepsilon_\delta + \varepsilon_2) \,\|\, p_\theta(z \mid x)) \right] \tag{53}$$

$$\geq \mathbb{E}_{\varepsilon_\delta} \min_{f \in \mathcal{F}} \mathbb{E}_{\varepsilon_2} \mathbb{E}_{\hat{p}(x)} \left[ D(f(x + \varepsilon_\delta + \varepsilon_2) \,\|\, p_\theta(z \mid x)) \right]. \tag{54}$$

Since $\mathcal{F}$ is closed under input translation,

$$\mathbb{E}_{\varepsilon_\delta} \min_{f \in \mathcal{F}} \mathbb{E}_{\varepsilon_2} \mathbb{E}_{\hat{p}(x)} \left[ D(f(x + \varepsilon_\delta + \varepsilon_2) \parallel p_\theta(z \mid x)) \right] = R(\theta \, ; \varepsilon_2). \tag{55}$$

It thus follows that $R(\theta \, ; \sigma_1) \geq R(\theta \, ; \sigma_2)$ for all $\theta \in \Theta$. $\qquad\square$

**Proposition 2.** *Let $\mathcal{P}$ be an exponential family with corresponding mean parameter space $\mathcal{M}$ and sufficient statistic function $T(\cdot)$. With a slight abuse of notation, consider $g \in \mathcal{G} : \mathcal{Z} \to \mathcal{M}$. Define $q(x, z) = \hat{p}(x) q(z \mid x)$, where $q(z \mid x)$ is a fixed inference model. Supposing $\mathcal{G}$ has infinite capacity, then the optimal generative model in Eq. (5) returns $g^*(z) = \mu \in \mathcal{M}$, where*

$$\mu = \sum_{i=1}^n q(x^{(i)} \mid z) \cdot T(x^{(i)}) = \sum_{i=1}^n \left( \frac{q(z \mid x^{(i)})}{\sum_j q(z \mid x^{(j)})} \cdot T(x^{(i)}) \right). \tag{19}$$

*Proof.* For a given inference model $q(z \mid x)$, the optimal generator maximizes the objective

$$\max_{g \in \mathcal{G}} \mathbb{E}_{\hat{p}(x)} \mathbb{E}_{q(z|x)} \left[ \ln g(z)(x) \right] = \max_{g \in \mathcal{G}} \mathbb{E}_{q(x,z)} \left[ \ln g(z)(x) \right]. \tag{56}$$

$$= \max_{g \in \mathcal{G}} \mathbb{E}_{q(x,z)} \left[ \ln p_{g(z)}(x) \right] \tag{57}$$

$$\leq \mathbb{E}_{q(z)} \max_{p \in \mathcal{P}} \mathbb{E}_{q(x|z)} \ln p(x) \tag{58}$$

$$= \mathbb{E}_{q(z)} \max_{\mu \in \mathcal{M}} \mathbb{E}_{q(x|z)} \ln p_\mu(x), \tag{59}$$

where $p_\mu$ denotes the distribution $p \in \mathcal{P}$ with associate mean parameter $\mu$. This inequality is tight since we can select $g^* \in \mathcal{G}$ such that

$$g^*(z) = \arg\max_{\mu \in \mathcal{M}} \mathbb{E}_{q(x|z)} \ln p_\mu(x). \tag{60}$$

Recall that the maximum likelihood and maximum entropy solutions are equivalent for an exponential family. From the moment-matching condition of maximum entropy, it follows that

$$g^*(z) = \arg\max_{\mu \in \mathcal{M}} \mathbb{E}_{q(x|z)} \ln p_\mu(x) \tag{61}$$

$$= \mathbb{E}_{q(x|z)} \left[ T(x) \right] \tag{62}$$

$$= \sum_{i=1}^n q(x^{(i)} \mid z) \cdot T(x^{(i)}) \tag{63}$$

$$= \sum_{i=1}^n \left( \frac{q(z \mid x^{(i)})}{\sum_j q(z \mid x^{(j)})} \cdot T(x^{(i)}) \right). \tag{64}$$

$\qquad\square$

**Proposition 3.** *Consider the regularizer $R_k(\theta \, ; \sigma, \mathcal{F}_H)$. Under similar assumptions as Proposition 1, then $R_{k_1}$ is stronger than $R_{k_2}$ when $k_1 \leq k_2$; i.e., $\min_\theta R_{k_1}(\theta \, ; \sigma, \mathcal{F}_H) = \min_\theta R_{k_2}(\theta \, ; \sigma, \mathcal{F}_H) = 0$ and $R_{k_1}(\theta \, ; \sigma, \mathcal{F}_H) \leq R_{k_2}(\theta \, ; \sigma, \mathcal{F}_H)$ for all $\theta \in \Theta$.*

*Proof.* Proof is provided in two parts.

**Part 1.** The relevant assumptions are that there exists $\theta \in \Theta$ such that $p_\theta(x, z) = p_\theta(z) p_\theta(x)$, and $f \in \mathcal{F}_H$ such that $f(x) = p_\theta(z)$. Note that $R_k$ is always non-negative. It follows readily that $\min_\theta R_k(\theta \, ; \sigma, \mathcal{F}_H) = 0$ for any choice of $k$.

**Part 2.** We define $\mathcal{L}_k$ as

$$\mathcal{L}_k = \mathbb{E}_{\hat{p}(x)} \ln p_\theta(x) - R_k(\theta \, ; \sigma, \mathcal{F}_H) \tag{65}$$

$$= \max_{f \in \mathcal{F}_H} \mathbb{E}_{\hat{p}(x)} \mathbb{E}_\varepsilon \mathbb{E}_{z_1 \dots z_k \sim f(x+\varepsilon)} \left[ \ln \frac{1}{k} \sum_{i=1}^k \frac{p_\theta(x, z_i)}{f(x + \varepsilon)(z_i)} \right]. \tag{66}$$

It suffices to show that $\mathcal{L}_k \geq \mathcal{L}_m$ when $k \geq m$. We adapt the proof from [8]'s Theorem 1 as follows. Let $|I| = m$ denote a uniformly distributed subset of distinct indices from $\{1, \ldots, k\}$. For any choice of $f \in \mathcal{F}_H$, the following inequality holds

$$\mathbb{E}_{\hat{p}(x)} \mathbb{E}_{\varepsilon} \mathbb{E}_{z_1 \ldots z_k \sim f(x+\varepsilon)} \left[ \ln \frac{1}{k} \sum_{i=1}^{k} \frac{p_\theta(x, z_i)}{f(x+\varepsilon)(z_i)} \right] \tag{67}$$

$$= \mathbb{E}_{\hat{p}(x)} \mathbb{E}_{\varepsilon} \mathbb{E}_{z_1 \ldots z_k \sim f(x+\varepsilon)} \left[ \ln \mathbb{E}_{I = \{i_1 \ldots i_m\}} \left[ \frac{1}{m} \sum_{j=1}^{m} \frac{p_\theta(x, z_{i_j})}{f(x+\varepsilon)(z_{i_j})} \right] \right] \tag{68}$$

$$\geq \mathbb{E}_{\hat{p}(x)} \mathbb{E}_{\varepsilon} \mathbb{E}_{z_1 \ldots z_k \sim f(x+\varepsilon)} \mathbb{E}_{I = \{i_1 \ldots i_m\}} \left[ \ln \frac{1}{m} \sum_{j=1}^{m} \frac{p_\theta(x, z_{i_j})}{f(x+\varepsilon)(z_{i_j})} \right] \tag{69}$$

$$= \mathbb{E}_{\hat{p}(x)} \mathbb{E}_{\varepsilon} \mathbb{E}_{z_1 \ldots z_m \sim f(x+\varepsilon)} \left[ \ln \frac{1}{m} \sum_{i=1}^{m} \frac{p_\theta(x, z_i)}{f(x+\varepsilon)(z_i))} \right] . \tag{70}$$

It thus follows that $\mathcal{L}_k \geq \mathcal{L}_m$. $\qquad \square$