[Reviews · NeurIPS 2018]

Reviewer 1



This paper puts forward the idea that we should in certain cases regularize the generative model in VAEs in order to improve generalization properties. Since VAEs perform maximum likelihood estimation, they can in principle exhibit the same overfitting problems as any other maximum likelihood model. This paper argues that we can regularize the generative model by increasing the smoothness of the inference model. The authors consider the Denoising VAE (DVAE) as a means of achieving such regularization. In the special case where the encoder is an exponential family, they show that the optimum natural parameters for any input data can be expressed as a weighted average over the optimum parameters for the data in the training set. Simlarly for a decoder model in the exponential family, the optimum choice of mean parameters for any latent code is a weighted average over sufficient statistics of training data. Experiments consider evaluate the test set log marginal likelihood for MNIST, Omniglot, and Caltech 101 with the DVAE, a convex combination between DVAE and VAE, and a third smoothing strategy based on regularization that bounds the magnitude of encoder weights. I think that this is a well-written paper that provides some useful insights about the interplay between encoder and decoder expressivity in VAE. Equations (14), (15) and (19) in particular provide a really nice intuition for the behavior of optimal encoders and decoders. The experiments are reasonable. I was not able to find anything amiss with the proofs (albeit on a cursory read). My only complaint about this paper is that there are some issues of clarity in the discussion of experiments approved (see questions below). Overall, this seems like a welcome addition to ongoing efforts to better understand the generalization properties of VAEs. I would be happy for this paper to appear. Comments and Questions - I'm not entirely sure what to make of the results in Table 1. While it seems intuitive that regularizing the inference model would improve the test set log marginal likelihood, it is less intuitive that doing so would also decrease Δ_inf, which one would naively expect to *increase*. Could the authors comment on this. - I was not quite able to follow the definition of the regularization parameter. This should probably be discussed in the main text, rather than the appendix. - When the authors write "Our main experimental contribution in this section is the verification that increasing the number of importance samples results in less underfitting when the inference model is over-regularized." It sounds a bit like they are saying that increasing the number of IWAE sample decreases susceptibility to underfitting. Is this in fact what they mean to say, or do these results simply validate Proposition 3? - While perhaps not the point per se of this paper, equation (19) provides some intuitions that are probably worth pointing out in expliclity. If we apply (19) to a standard VAE with Bernoulli likelihood, then the mean value of each pixel will be a weighted average over pixel values of examples in the training data encode to similar z values. My read of this is that, for an optimal decoder, "interpolation" really means "averaging over nearest neighbors in the latent space". - For a normal VAE, the mutual information I(x;z) in the inference model tends to saturate to log n (where n is the size of the training data). This suggests that the weights q(x^(i) | z) would be pretty close to 0 for all (i) except the nearest neighbor for a normal VAE (since each x^(i) encode to non-overlapping regions in z space). In other words, for a normal VAE, an optimal decoder could effectively be doing nearest-neighbor matching in the latent space. As you increase the regularization, presumably the generative model averages over a larger number of neighbors. It would be interesting to look at how the expected entropy of q(x^(i) | z) changes with the amount of regularization (the exponent of this entropy would effectively give you something like "the number of neighors" that an optimal likelihood model would average over). - The posteriors in Figure 2 of cited ref [10] look pretty significantly non-Gaussian to me. I unfortunately lacked time to read ref [10] in detail, but I was not immediately able to find support for the claim that "it has still been observed empirically that the true posteriors tend to be approximately Gaussian by the end of training as a result of posterior regularization induced by the variational family". Could the authors either clarify, or walk back this claim? - The notation F(Q) does not make sense to me. In this notation would the domain of F somehow be the set of all possible variational families? Given that Q is a specific family of variational distributions (in practice a parametric family like a Gaussian with diagonal covariance), and F(Q) is a family of functions that map from X to Q, it is not clear how you would even apply the same F to two different families Q and Q'. Writing F_Q, or perhaps just F would be clearer, since F is in practice specific to a family Q. The same argument applies to G(P). - How are the errors in Table 1 defined? Is this an error over independent restarts? If so, how many restarts? Could the authors figure out a way to reformat the table so the erros are not shown in tiny fonts? I understand that we're all using screens to read papers these days, but I still think that fonts should be legible in print. - Does Table 1 show results for the regularization parameters listed in Table 7? - Do Figures 1 a 2 show the same IW-SVI approximation of log p_θ(x) as Table 1? - Why does the Caltech panel of Figure 1 have error bars, whereas the Omniglot and MNIST panels do not? - q(x^(i) | z) is never explicitly defined, as far as I can tell. - The acronym "AIR" results in an unfortunate collision with "Attend, Infer, Repeat", which at this point is pretty well known.

Reviewer 2



In this paper, the authors address the 'amortization gap' introduced in recent work and propose means to regularize inference networks in order to ameliorate its effects. The amortization gap is an effect where bottlenecks in amortized inference networks lead to both suboptimal inference as well as suboptimal generation due to the fact that the generator can never be fit adequately. Specifically, the authors propose a fix by injecting noise into the inference networks by adding it to the observations that are conditioned upon. The authors also derive a loss function for this regularizer R which can be combined with variational inference. Quality: The paper has high quality in its style as it blends intuitions with formal proofs for its conceptually simple key idea. I particularly enjoyed the link to importance weighted AEs and the clear explanation that in that case the regularization term will not work as well in high sample sizes, since the variational family becomes very general as is. The key idea of the paper is intriguing, but causes some problems from the modeler's point of view: 1. what is the distribution of the noise that one should add to the inference network inputs? 2. What is the link of that noise to the generator noise? Can we posit a model of the perturbation for the inputs to learn those? 3. Is the addition of Gaussian noise the best we can do or just convenient for proof-reasons? We already know that Gaussian noise is a bad observation model for image patches with VAEs, why would it be the 'right' regularizer for inference? The paper inspires such questions but offers little in terms of exploration of these topics, which ultimately may lead to more useful general regularizers. Experimentally the paper is limited, but does a convincing job of showing the effects it discusses in practical settings. A main question that remains is the following: where is the tradeoff between regularizing inference networks and defining better variational families through inference algorithms? We already know that we can get over the limits posed here by various methods blending VI and sampling techniques, is the proposed alternative a scalable way out in the opinion of the authors? Clarity: - The paper makes lucid points and has a pleasant style by formally presenting motivations and then proofs in its main text in clear order. While this may be a burden to the casual reader, it is a very complete story. -It is unclear in the flow of the paper what role Section 3.2 plays as it does not follow from the previous ones It seems to not be necessary for the key points of the paper since it is an almost orthogonal idea to the main idea but additionally also is not presented at length. It is intuitively understandable that smooth weights will have regularizing effects, but this aspect of inference regularization might benefit from a more thorough treatment in future work. Originality: The paper follows closely in the vein of the cited work "Inference Suboptimality In Variational Autoencoders", but adds a potential solution to the literature. It is as such appropriately novel for a nips submission, since the cited work is also very recent. Significance: The topic of regularizing amortized inference machines is timely and important, as these objects drive a lot of modern probabilistic Deep Learning. I have doubts that the proposed solution will be sufficiently general to be widely adopted, but it is a good formal step. UPDATE Post Rebuttal: raising score to 7.

Reviewer 3



Thank you for an interesting read. The authors consider over-fitting issues of latent variable model training and propose regularising the encoder for better generalisation. Two existing regularisation techniques -- denoising VAE and weight normalisation for encoder parameters -- are analysed in both theoretical and empirical way. The denoising VAE regularisation mechanism is further extended to IWAE. I think this is an interesting paper, which explicitly point out the folklore wisdom that the encoder in VAE introduces bias for the generator training. The analysis also has some interesting point. However there are a few concerns that needs to be addressed, which drives me away from recommending strong acceptance. I will be happy to read through the authors' feedback and see if I need to adjust my score. 1. training procedure of p and q: VAE can be viewed as a coupled update process of variational EM, which trains q with variational inference, and uses q in the M-step for training p. But generally speaking, it is unnecessary to tie the objective of training p and q: I can train p with E_q(z|x)[\log p(x, z)], but train q(z|x) with what-ever method that make sense to me. So I think there are several questions here: (1a) Do you want to be Bayesian for inference? If no then you can definitely claim the denoising VAE as some other inference method that make sense. (1b) Should I treat q(z|x) as part of my model or not, since I will use q(z|x) to compute all the statistics for test anyway? If yes then it is very natural to motivate regularisations on q. But people who think the answer is no will go for direct regularisation of the p model, and I'm confident that they will be able to figure out a way. I would be happy to see your answers on these questions. This would help me a lot to better understand the scope of your proposed research. 2. Below your definition 1, what do you mean by "scalar addition to share the same minimum"? 3. In your proposition 1, if their exist a model p that factorises, then this means p can ignore z, do you think your method will push p towards this solution when sigma is large? Consider sigma tends to infinity for example. 4. In Table 1, is that possible to report ESS for your estimate? Just for sanity check, if the ESS are roughly the same across evaluated models then your results are more reliable. 5. In the discussion of Proposition 2, you mentioned as the smoothness increases mu tends to an empirical average of sufficient statistics, meaning that p is the maximum entropy solution. Can you discuss why maximum entropy solution might be something you want for better generalisation? 6. Your generalisation of denoising VAE to IWAE: does the \tilde{D} definition always non-negative? Why not directly start from a proper definition of unnormalised KL? I had this question already when I saw [22] quite a while ago... 7. You reported inference gap in your tables in the main text. I'm not exactly sure what you want to demonstrate here. From my understanding of appendix C you trained another q(z|x) on test data, and I guess you used the same q(z|x) to estimate log p(x) and L(x) (or L_8(x))? Then in principle I don't need to use the same family of q distribution in training/test time, and the inference gap can be reduced to arbitrary small if the test-time q(z|x) is super powerful. ========================================== I have read the feedback and engaged in discussions with other reviewers. My rating stays the same, and I hope the authors could better motivate the purpose of the paper (e.g. discuss more on other regularisation methods say beta-VAE and discuss what they can achieve). I feel the inference gap results are still confusing. Also better to have a discussion on how the quality of amortised inference relates to generalisation.